# Hyaluronic acid fuels pancreatic cancer cell growth

Peter K Kim[1,2†], Christopher J Halbrook[2†], Samuel A Kerk[1,2†], Megan Radyk[2], Stephanie Wisner[2], Daniel M Kremer[2,3], Peter Sajjakulnukit[1,2], Anthony Andren[2], Sean W Hou[2], Ayush Trivedi[2], Galloway Thurston[2], Abhinav Anand[2], Liang Yan[4], Lucia Salamanca-Cardona[5], Samuel D Welling[2], Li Zhang[2], Matthew R Pratt[6,7], Kayvan R Keshari[5,8], Haoqiang Ying[4], Costas A Lyssiotis[2,9,10]*

[1]Doctoral Program in Cancer Biology, University of Michigan, Ann Arbor, United States; [2]Department of Molecular & Integrative Physiology, University of Michigan, Ann Arbor, United States; [3]Program in Chemical Biology, University of Michigan, Ann Arbor, United States; [4]Department of Molecular and Cellular Oncology, University of Texas MD Anderson Cancer Center, Houston, United States; [5]Department of Radiology, Memorial Sloan Kettering Cancer Center, New York City, United States; [6]Department of Chemistry, University of Southern California, Los Angeles, United States; [7]Department of Biological Sciences, University of Southern California, Los Angeles, United States; [8]Molecular Pharmacology Program, Memorial Sloan Kettering Cancer Center, New York City, United States; [9]Department of Internal Medicine, Division of Gastroenterology and Hepatology, University of Michigan, Ann Arbor, United States; [10]Rogel Cancer Center, University of Michigan, Ann Arbor, United States

*For correspondence: clyssiot@med.umich.edu

†These authors contributed equally to this work

**Abstract** Rewired metabolism is a hallmark of pancreatic ductal adenocarcinomas (PDA). Previously, we demonstrated that PDA cells enhance glycosylation precursor biogenesis through the hexosamine biosynthetic pathway (HBP) via activation of the rate limiting enzyme, glutamine-fructose 6-phosphate amidotransferase 1 (GFAT1). Here, we genetically ablated GFAT1 in human PDA cell lines, which completely blocked proliferation in vitro and led to cell death. In contrast, GFAT1 knockout did not preclude the growth of human tumor xenografts in mice, suggesting that cancer cells can maintain fidelity of glycosylation precursor pools by scavenging nutrients from the tumor microenvironment. We found that hyaluronic acid (HA), an abundant carbohydrate polymer in pancreatic tumors composed of repeating N-acetyl-glucosamine (GlcNAc) and glucuronic acid sugars, can bypass GFAT1 to refuel the HBP via the GlcNAc salvage pathway. Together, these data show HA can serve as a nutrient fueling PDA metabolism beyond its previously appreciated structural and signaling roles.

## Editor's evaluation

In this manuscript, the authors report that a major component of the pancreatic cancer microenvironment, hyaluronic acid, provides an important source of metabolic intermediates required for pancreatic cancer growth during conditions of nutrient limitation. Overall, this work nicely combines multiple cell lines and genetic approaches to show that scavenging of N-acetyl-glucosamine from hyaluronic acid supports cancer cell growth when de novo synthesis of N-acetyl-glucosamine is limited. More broadly, this work highlights how reliance on certain metabolic enzymes can vary depending on whether cells are grown in vitro or in vivo and provides evidence that environmental nutrient scavenging can contribute to differential metabolic dependencies of cancer cells.

## Introduction

Pancreatic ductal adenocarcinoma (PDA) is one of the deadliest human cancers with no clinically effective treatment options (*Siegel et al., 2020*). PDA is characterized by an intense fibroinflammatory stroma, poor vascularity, deregulated nutrient levels, and rich deposition of extracellular matrix (ECM) components. To survive and proliferate in this nutrient austere tumor microenvironment (TME), the oncogenic driver in PDA, mutant Kras, facilitates the rewiring of PDA metabolism (*Halbrook and Lyssiotis, 2017*; *Perera and Bardeesy, 2015*; *Ying et al., 2016*; *Kerk et al., 2021*).

One example of this, which we have demonstrated in previous work (*Ying et al., 2012*), is the effect of mutant Kras on the activity of the hexosamine biosynthetic pathway (HBP). Signaling downstream of Kras results in upregulated expression of *Gfpt1*, which encodes glutamine-fructose 6-phosphate amidotransferase 1 (GFAT1). The HBP is an evolutionarily conserved pathway that integrates glucose, glutamine, fatty acid, and nucleotide metabolism to generate the final product uridine diphosphate *N*-acetylglucosamine (UDP-GlcNAc). UDP-GlcNAc is a crucial donor molecule for glycosylation and *O*-GlcNAcylation, two essential post-translational modifications required for cellular structure, signaling, and survival (*Akella et al., 2019*). The HBP is the only pathway able to generate UDP-GlcNAc de novo. Because the HBP integrates nutrients from several major macromolecular classes to produce UDP-GlcNAc, it predictably functions as a nutrient sensing mechanism for available energy within a cell (*Wellen et al., 2010*). Indeed, numerous studies across cancer subtypes have demonstrated how HBP activity is enhanced to support tumor survival and growth (*Paszek et al., 2014*; *Akella et al., 2019*; *Walter et al., 2020*; *Yang et al., 2016*) and even immune evasion through alteration of extracellular glycosylation content (*Lee et al., 2019*).

A compendium of studies during the last decade have revealed that PDA cells fuel their aberrant metabolic programs through nutrient scavenging (*Ying et al., 2012*; *Commisso et al., 2013*; *Kamphorst et al., 2015*; *Yang et al., 2011*; *Davidson et al., 2017*; *Olivares et al., 2017*). Mechanisms include sustained activation of intracellular recycling pathways (e.g. autophagy), the upregulation of nutrient transporter expression (e.g. carbohydrate, lipid, and amino acid transporters), and the activation of extracellular nutrient scavenging pathways (e.g. macropinocytosis). Further, PDA cells also participate in metabolic crosstalk and nutrient acquisition with non-cancerous cells in the TME, such as cancer-associated fibroblasts (CAFs) and tumor-associated macrophages (TAMs) (*Sousa et al., 2016*; *Dalin et al., 2019*; *Halbrook et al., 2019*; *Zhu et al., 2020*; *Lyssiotis and Kimmelman, 2017*). A notable example is the observation that PDA cells can directly obtain nutrients from the CAF-derived ECM, such as collagen (*Olivares et al., 2017*). Taken together, elucidating the interaction of PDA cells with different cell populations and ECM components will be instrumental for delineating deregulated PDA metabolism and improving therapeutic strategies.

A major structural component of the TME is hyaluronic acid (HA), a hydrophilic glycosaminoglycan. HA is ubiquitously present in human tissue, especially in skin, connective tissue, and joints, and it is richly abundant in pancreatic tumors (*Theocharis et al., 2000*). HA is thought to be primarily deposited by CAFs and, to some extent, by PDA cells (*Goossens et al., 2019*; *Mahlbacher et al., 1992*). HA avidly retains water, which is responsible for both its lubricating properties and, in PDA tumors, the supraphysiological pressure that impairs vascularity and limits drug penetrance (*Jacobetz et al., 2013*; *Provenzano et al., 2012*). An aspect of HA biology that has not previously been studied is its potential role as a nutrient. This is surprising given that HA is a carbohydrate polymer whose monomeric unit is a disaccharide of glucuronic acid and *N*-acetyl-glucosamine (GlcNAc).

Herein, we set forth to determine the utility of targeting the HBP in PDA. We found that GFAT1 was required for cell survival in vitro. In marked contrast, GFAT1 knockout tumors readily grew in vivo. Based on this observation, we hypothesized that GlcNAc-containing components of the TME could bypass the HBP in vivo by way of the GlcNAc salvage pathway. We demonstrate that HA can be metabolized by PDA cells to support survival and proliferation by refilling the HBP. In sum, our study identifies HA as a novel nutrient source in PDA and contributes to a growing body of data illuminating the important role of the TME in cancer metabolism.

# Results

## Pancreatic cancer cells require de novo HBP fidelity in vitro but not in vivo

Previously, we found that mutant Kras transcriptionally activates *Gfpt1* expression downstream of MAPK signaling in a murine model of PDA to facilitate HBP activity (*Ying et al., 2012*). GFAT1 catalyzes the reaction that generates glucosamine 6-phosphate and glutamate from fructose 6-phosphate and glutamine (*Figure 1A*). In another previous study we demonstrated that PDA cells are dependent on glutamine anaplerosis for proliferation (*Son et al., 2013*). Thus, we hypothesized that inhibiting GFAT1 in PDA would have the simultaneous benefit of blocking two major metabolic pathways that support PDA proliferation, thereby providing a considerable therapeutic window.

Our previous results targeting GFAT1 in murine cells with shRNA yielded insufficient knockdown to draw a conclusion as to its necessity in PDA (*Ying et al., 2012*). Thus, here we used CRISPR/Cas9 to knockout GFAT1 from three established human PDA cell lines: HPAC, TU8988T, and MiaPaCa2. During selection, the pooled polyclonal populations were grown in GlcNAc, which bypasses GFAT1 via the GlcNAc salvage pathway (*Figure 1A*). This supplement was included to minimize metabolic rewiring within the selected populations.

The GFAT1 knockout lines had differential response to GlcNAc withdrawal. Among the three GFAT1 knockout cell lines, only the HPAC line exhibited a marked reduction in cell number, consistent with loss of viability, in the 4 days following GlcNAc withdrawal (*Figure 1—figure supplement 1A*). The impact on proliferation was similarly reflected in decrease of the UDP-GlcNAc pool, which was analyzed using liquid chromatography-coupled tandem mass spectrometry (LC-MS/MS) (*Figure 1—figure supplement 1B*). Consistent with the proliferative phenotypes across lines, the HPAC line also had a significantly smaller UDP-GlcNAc pool than that of either MiaPaCa2 or 8988T cells (*Figure 1—figure supplement 1C*). Cellular *O*-GlcNAcylation of the proteome was also measured by immunoblot 3 days after GlcNAc withdrawal. Again, consistent with the LC-MS/MS analysis, *O*-GlcNAc expression was significantly reduced in HPAC but was maintained in TU8988T (*Figure 1—figure supplement 1D*).

The data from TU8988T and MiaPaca2 were similar to those from our earlier studies (*Ying et al., 2012*), and thus we posited that knockout of GFAT1 was incomplete. As such, we subsequently generated clonal cell lines from the pooled populations. This analysis revealed that the degree of GFAT1 knockout varied by cell line and by clone, and this correlated with differential growth and sensitivity to GlcNAc withdrawal in vitro (*Figure 1—figure supplement 1E, F*). Clones for each cell line without detectable GFAT1 expression (*Figure 1B*) were further validated by sequencing and were subsequently used to examine the role of the HBP without interference from GFAT1-proficient cells.

Using our genomically sequenced and bona fide GFAT1 knockout clonal lines, we found that GFAT1 knockout led to an abolishment of colony formation (*Figure 1C*) and potently impaired proliferation (*Figure 1D*, *Figure 1—figure supplement 1G*) in all three PDA cell lines in vitro. We then moved these cells into in vivo tumor models. Surprisingly, when either the pooled or the clonal knockout lines were implanted into the flanks of immunocompromised mice, they readily formed tumors that were comparable to their wildtype counterparts in terms of weight and volume (*Figure 1E and F* and *Figure 1—figure supplement 1H*). Similar results were obtained for GFAT1 knockout clonal lines implanted orthotopically into the pancreas (*Figure 1G*). Of note, while clearly capable of forming tumors, the GFAT1 knockout clonal lines grown in the pancreas were smaller than the wildtype tumors at endpoint. The marked discrepancy in phenotype between in vitro and in vivo settings led us to hypothesize that GFAT1 knockout clones were scavenging nutrients from the TME to refill the HBP, which enabled their survival and tumor growth.

## Conditioned media rescues proliferation of GFAT1 knockout PDA cells

To test our scavenging hypothesis, we generated conditioned media (CM) from CAFs, the most abundant stromal cell type in the pancreatic TME (*Zhang et al., 2019*; *Neesse et al., 2019*). When GFAT1 knockout clones were incubated in patient-derived CAF CM, we observed a significant, albeit modest, rescue in colony formation (*Figure 2A and B*). Unexpectedly, we observed a more robust, dose-dependent rescue of colony formation in GFAT1 knockout cells with CM from wildtype TU8988T cells (*Figure 2C–F* and *Figure 2—figure supplement 1A*). Similarly, CM from wildtype HPAC and MiaPaCa2

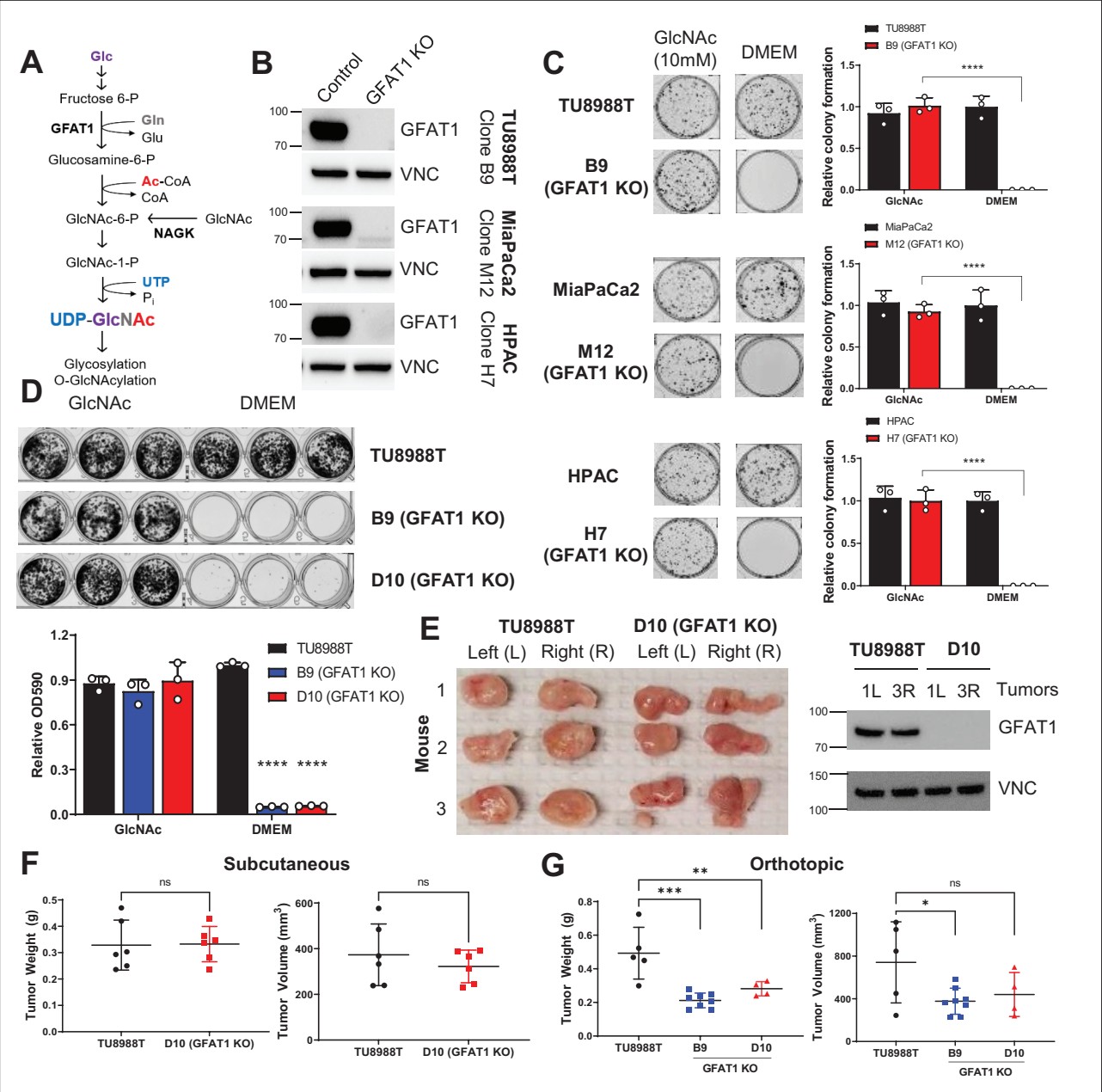

**Figure 1.** Pancreatic ductal adenocarcinoma (PDA) requires de novo hexosamine biosynthetic pathway (HBP) fidelity in vitro but not in vivo.
(**A**) Schematic overview of the HBP and the nutrient inputs. Ac-CoA, acetyl-coenzyme A; GFAT1, glutamine fructose 6-phosphate amidotransferase 1; Glc, glucose; GlcNAc, *N*-acetyl-glucosamine; Gln, glutamine; Glu, glutamate; NAGK, *N*-acetyl-glucosamine kinase; $P_i$, inorganic phosphorus; UTP, uridine-triphosphate. (**B**) Western blot of GFAT1 and loading control VINCULIN (VNC) from validated CRISPR/Cas9 knockout TU8988T, MiaPaca2, and HPAC clones and their control (non-targeted sgRNA). (**C**) Representative wells from a colony-forming assay in parental and clonally derived GFAT1 knockout cell lines grown in base media (DMEM) or base media supplemented with 10 mM GlcNAc. Data quantitated at right, n = 3. (**D**) Proliferation assay in parental and two GFAT1 knockout clonal TU8988T cell lines. Representative wells are presented above data quantitated by crystal violet extraction and measured by optical density (OD) at 590 nm, n = 3. (**E**) Tumors from parental TU8988T (n = 6) and GFAT1 knockout clone D10 (n = 6) grown subcutaneously in immunocompromised mice. Accompanying western blot for GFAT1 and VNC loading control from representative tumor lysates. (**F**) Tumor volume and tumor weight from samples in **E**. (**G**) Tumor volume and tumor weight from parental TU8988T (n = 5) and GFAT1 knockout clones B9 (n = 8) and D10 (n = 4) implanted and grown orthotopically in the pancreas of immunocompromised mice. Error bars represent mean ± SD. n.s., non-significant; *p < 0.05; **p < 0.01; ***p < 0.001; ****p < 0.0001.

The online version of this article includes the following figure supplement(s) for figure 1:

**Figure supplement 1.** Additional characterization of GFAT1 knockout pancreatic ductal adenocarcinoma (PDA) populations and clonal lines.

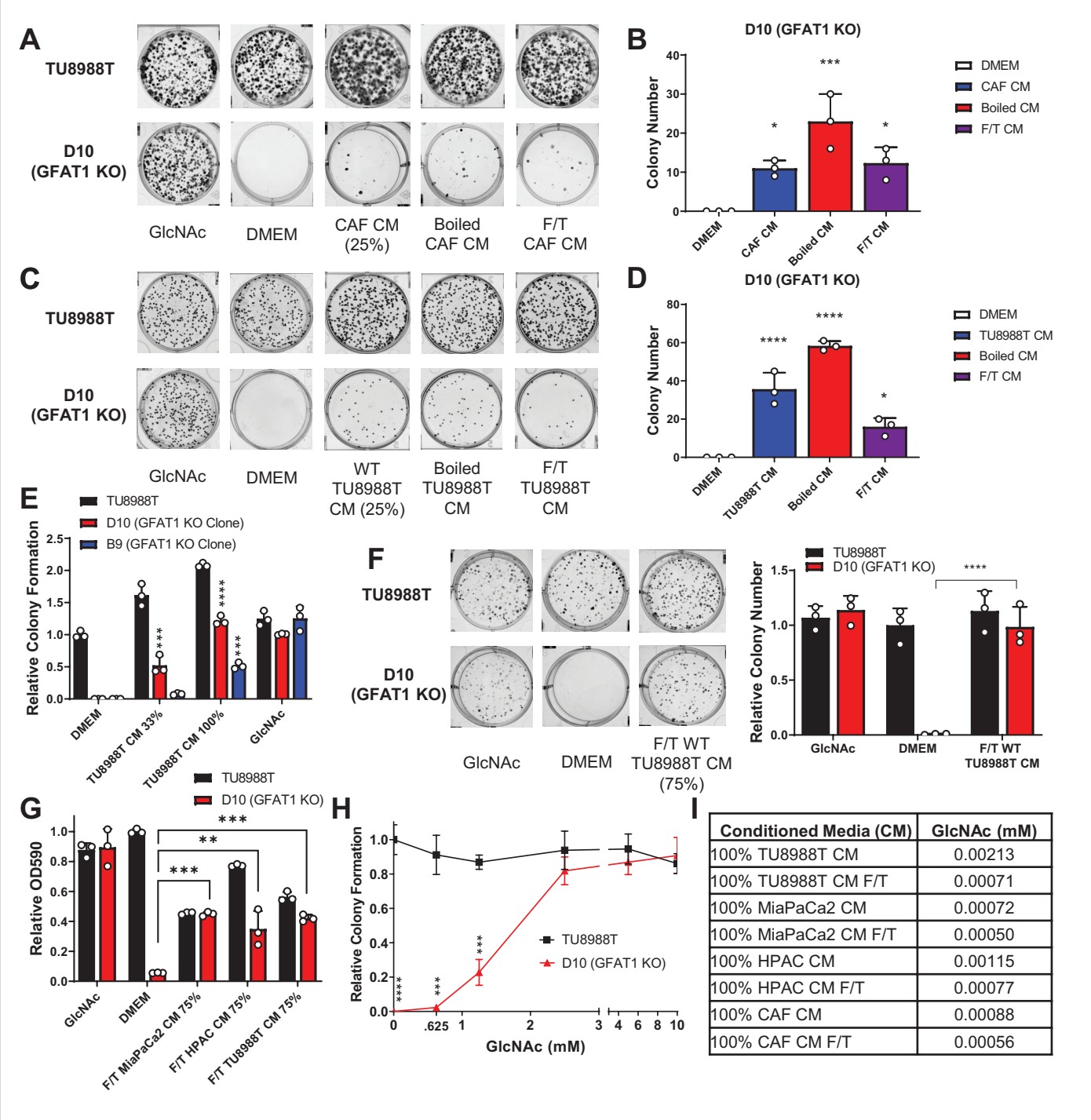

**Figure 2.** Conditioned media (CM) from cancer-associated fibroblasts (CAFs) and wild-type pancreatic ductal adenocarcinoma (PDA) cells support proliferation of GFAT1 knockout cells. (**A**) Representative wells from a colony-forming assay in parental TU8988T and GFAT1 knockout clonal line D10 in 10 mM *N*-acetyl-glucosamine (GlcNAc), base media (DMEM), or base media supplemented 1:3 (25%) with CAF CM, boiled CAF CM, or CAF CM subject to freeze-thaw (F/T). (**B**) Quantitation of colonies from data in **A** (n = 3). (**C**) Representative wells from a colony-forming assay in parental TU8988T and GFAT1 knockout clonal line D10 in 10 mM GlcNAc, DMEM, or base media supplemented 1:3 (25%) with CM from wildtype TU8988T cells, boiled TU8988T CM, or TU8988T CM subject to F/T. (**D**) Quantitation of colonies from data in **C** (n = 3). (**E**) Quantitation of colony-forming assay data of parental and GFAT1 knockout clonal TU8988T lines in base media, positive control GlcNAc, wildtype TU8988T CM diluted 1:2 (33%) or used directly (100%) (n = 3). (**F**) Representative wells and quantitation of colony-forming assay data of parental and GFAT1 knockout clonal TU8988T lines in base media, positive control GlcNAc, and wildtype TU8988T CM subject to F/T and diluted 3:1 (75%) (n = 3). (**G**) Quantitation of colony-forming assay data of parental and GFAT1 knockout clonal TU8988T lines in base media, positive control GlcNAc, or wildtype TU8988T, HPAC, or MiaPaCa2 CM subject to F/T

*Figure 2 continued on next page*

*Figure 2 continued*

and diluted 3:1 (75%) (n = 3). (**H**) GlcNAc dose response curve presented as relative colony number for parental and GFAT1 knockout TU8988T cells (n = 3). (**I**) Mean of absolute quantitation of GlcNAc in various CM by liquid chromatography-coupled tandem mass spectrometry (LC-MS/MS) (n = 3). Error bars represent mean ± SD. *p < 0.05; **p < 0.01; ***p < 0.001; ****p < 0.0001.

The online version of this article includes the following figure supplement(s) for figure 2:

**Figure supplement 1.** Rescue activity of conditioned media (CM) and *N*-acetyl-glucosamine (GlcNAc) in GFAT1 knockout cells.

cells was able to partially rescue proliferation of a subset of GFAT1 knockout clones (*Figure 2G* and *Figure 2—figure supplement 1B, C*).

To begin to identify the rescue factors in the CM, we subjected the CM to boiling or repeated cycles of freezing and thawing (F/T). In each of these conditions, both the CAF and the PDA CM retained the ability to support colony formation in GFAT1 knockout cells (*Figure 2A–D*). These results suggested the relevant factor(s) did not require tertiary structure. Additionally, we observed that the rescue activity of the CM was dose dependent (*Figure 2E–G* and *Figure 2—figure supplement 1A-C*).

As GlcNAc was used to establish our GFAT1 knockout lines, we first quantitated the GlcNAc concentration in the CM by mass spectrometry. GlcNAc dose response curves demonstrated that millimolar quantities of GlcNAc (>0.625 mM) were required to rescue colony formation of GFAT1 knockout PDA cells (*Figure 2H* and *Figure 2—figure supplement 1D*). By contrast, LC-MS/MS quantification of GlcNAc in the CM revealed that it was in the low micromolar range (*Figure 2I*), several orders of magnitude below the millimolar doses of exogenous GlcNAc required to maintain proliferation (*Figure 2H* and *Figure 2—figure supplement 1D*). These results illustrated that free GlcNAc was not the relevant molecule in the CM mediating rescue. This led us to consider alternate possibilities, including GlcNAc-containing components of the TME.

## HA rescues GFAT1 knockout PDA cells

GlcNAc is a widely utilized molecule as a structural component of the ECM, a modification of various lipid species, and a post-translational modification on proteins (*Moussian, 2008*; *Bond and Hanover, 2015*). Thus, we hypothesized that GlcNAc was released into CM as a component part of a lipid, protein, or glycosaminoglycan polymer, and that this mediated rescue of *GFAT1* knockout. To test this, we first applied necrotic cellular debris from the murine hematopoietic FL5.12 cell line (*Kim et al., 2018*) to GFAT1 knockout cells grown at clonal density. The necrotic cellular debris contains the full complement of biomolecules, including GlcNAc-containing proteins and lipids. Necrotic cell debris was unable to rescue GFAT1 knockout across our cell line panel (*Figure 3—figure supplement 1A-F*). Next, we tested if glycosaminoglycan carbohydrate polymers could mediate rescue of GFAT1 knockout, in a matter akin to CM. High-dose heparin was not able to rescue colony formation in GFAT1 knockout cells (*Figure 3—figure supplement 1A-F*). In contrast, 78 kDa HA provided a modest but significant rescue (*Figure 3A and B*).

HA is a carbohydrate polymer and an ECM component that is abundant in the PDA TME (*Theocharis et al., 2000*). The monomeric form of HA is a repeating disaccharide consisting of glucuronic acid and GlcNAc. HA polymer length, often described by its molecular weight (MW), has important impacts on its biological activity. In non-pathological settings, newly synthesized HA is predominantly high molecular weight (HMW; >1000 kDa) (*Monslow et al., 2015*). However, in tumors and tumor interstitial fluid, there is a significantly elevated level of low molecular weight (LMW; 10–250 kDa) and oligo-HA (o-HA; <10 kDa) (*Schmaus et al., 2014*; *Lv et al., 2007*). Consistent with the rescue of colony formation in GFAT1 knockout cells, LMW HA (78 kDa) was also able to rescue total proteome *O*-GlcNAc levels, as assessed by western blot (*Figure 3C–E*).

Cancer cells have been reported to uptake HA via macropinocytosis (*Greyner et al., 2010*). Thus, a possible explanation for the modest rescue could be low macropinocytosis activity. However, in PDA, mutant Kras drives macropinocytosis (*Commisso et al., 2013*), and quantitation of macropinocytotic activity with a fluorescent dextran-based assay revealed that our three PDA cell lines exhibited considerable macropinocytosis (*Figure 3—figure supplement 1G*). Similarly, following HA uptake using a fluorescently conjugated HA (HA-FITC) revealed GFAT1-proficient and GFAT1 knockout cells readily take up HA (*Figure 3F*, *Figure 3—figure supplement 1H*).

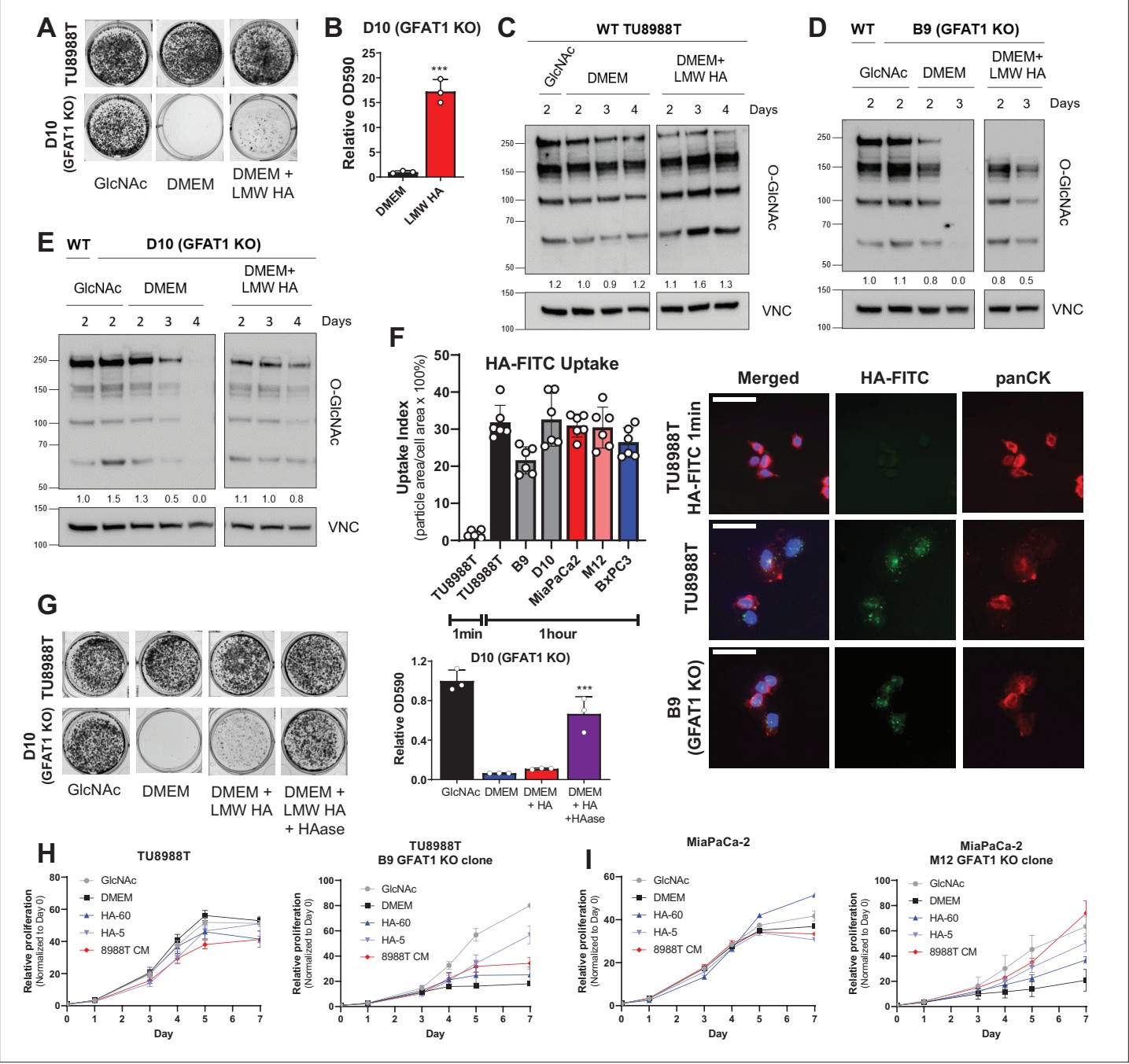

**Figure 3.** Hyaluronic acid rescues GFAT1 knockout pancreatic ductal adenocarcinoma (PDA) cells. (**A**) Representative wells from a colony-forming assay in parental and clonally derived GFAT1 knockout TU8988T cell lines grown in base media (DMEM), positive control *N*-acetyl-glucosamine (GlcNAc) (10 mM), or low molecular weight (LMW) hyaluronic acid (78 kDa HA, 10 mM). (**B**) Quantitation of data from **A** (n = 3). (**C**) Western blot of proteome *O*-GlcNAc and loading control VINCULIN (VNC) in parental TU8988T cells grown in base media (DMEM) plus GlcNAc or LMW HA for the indicated time points. Band density was quantitated, normalized to control, and presented below the blot. (**D,E**) Western blot of proteome *O*-GlcNAc and loading control VNC in GFAT1 knockout clonal lines (**D**) B9 and (**E**) D10 in base media (DMEM) plus GlcNAc or LMW HA for the indicated time points. Wildtype (WT) TU8988T included as control. Band density was quantitated, normalized to control, and presented below the blot. (**F**) Quantitation of HA-FITC uptake in wildtype (TU8988T, MiaPaCa2, BxPC3) and GFAT1 knockout clones (B9, D10, M12) presented as percent total particle area over total cell area at 1 min or 1 hr; n = 6 frames per condition. Cell area was calculated by staining for pan-cytokeratin (panCK). At right, representative images. Scale bar, 50 μm. (**G**) Representative wells of a proliferation assay in parental TU8988T and GFAT1 knockout clonal line D10 grown in base media (DMEM), positive control GlcNAc (10 mM), or base media supplemented 1:1 with boiled LMW HA (10 mM) with and without pre-digestion with hyaluronidase (HAase). At endpoint, cells are stained with crystal violet, and the stain was then extracted and quantitated by OD at 590 nm (n = 3). (**H,I**) Proliferation time course,

*Figure 3 continued on next page*

*Figure 3 continued*

as measured on the Incucyte, of (**H**) TU8988T and (**I**) MiaPaCa parental and GFAT1 knockout cells in base media (DMEM), positive control (GlcNAc), 60 kDa HA (LMW HA), 5 kDa HA (o-HA), or wildtype TU8988T CM (n = 3). Error bars represent mean ± SD. ***p < 0.001.

The online version of this article includes the following figure supplement(s) for figure 3:

**Figure supplement 1.** Characterization of macropinocytosis and glycosaminoglycan rescue activity in pancreatic ductal adenocarcinoma (PDA) and GFAT1 knockout cells.

**Figure supplement 2.** Analysis of hyaluronic acid formulation on GFAT1 rescue and composition in conditioned media (CM).

This led us to hypothesize that the rate limiting step is not HA entry into cells, but rather, the cleavage of HA into smaller fragments. Consistent with this hypothesis, utilizing hyaluronidase to break down LMW HA enhanced the rescue of colony formation (*Figure 3G*). Of note, hyaluronidase was heat-inactivated after it was used to cleave HA (i.e. before its application to GFAT1 knockout cells; *Figure 3—figure supplement 1I*), as hyaluronidase has been reported to directly impact cellular metabolism (*Sullivan et al., 2018*). Next, we tracked the rescue of proliferation using HA of varying size: LMW HA (60 kDa) and o-HA (5 kDa). This analysis revealed that HA-mediated rescue, as measured through proliferation quantification, was considerably higher for o-HA than for LMW HA, and comparable to PDA CM (*Figure 3H, I* and *Figure 3—figure supplement 2A-D*).

To relate these studies back to our CM rescue studies, we employed an enzyme-linked immunosorbent assay (ELISA) to quantitate HA in the PDA and CAF CM (*Figure 3—figure supplement 2E*). CAF CM contained considerably more HA than PDA CM, consistent with the known role for fibroblasts in the production of HA (*Figure 3—figure supplement 2F*). However, we found that the lower limit of detection for the ELISA was ~50 kDA HA. Indeed, we demonstrated this by digesting 10 mM HA, CAF CM, or PDA CM with HAase. Post-digestion, we are unable to detect appreciable levels of HA (*Figure 3—figure supplement 2F,G*). These results illustrate the CAF CM produces more HMW HA, which we posit explains its limited rescue of GFAT1 knockout in vitro.

HA is produced by the family of HA synthases, which can be inhibited in vitro with the tool compound 4-methylumbelliferone (4-MU) (*Nagy et al., 2015*). To determine if HA is the relevant metabolite in wildtype PDA CM facilitating rescue of GFAT1 knockout, we first treated wildtype PDA cells with a range of 4-MU doses. Cell proliferation was followed to assess off-target toxicity and HA in CM was quantitated using the HA ELISA. Of note, while PDA cells produce much less detectable HA than CAFs, they do produce a sufficient amount so as to analyze 4-MU activity (*Figure 3—figure supplement 2F*). Indeed, we demonstrated that 4-MU dose dependently blocked HA release into CM (*Figure 4A*) while also exhibiting proliferative defects at the highest concentration tested (*Figure 4B–D*). Application of CM from 4-MU-treated wildtype PDA cells in GFAT1 rescue assays revealed that the reduction in HA content paralleled the decrease in CM rescue activity in GFAT1 knockout cells (*Figure 4E*). These collective in vitro studies illustrate that HA can rescue the proliferation of GFAT1 knockout cells and strongly suggest that HA is the relevant factor mediating the rescue activity of PDA CM. However, while we believe these collective results are convincing, detection and quantitation of o-HA in PDA CM will be required to draw a definitive conclusion.

To relate this work back to our tumor studies (*Figure 1E–G*), we probed for HA content in tumor slices by immunohistochemical detection of HA binding protein (HABP) (*Provenzano et al., 2012*). We stained normal pancreas as the negative control (note: positive staining of vasculature) and a murine pancreatic tumor as a positive control (*Figure 4F*). We then stained 10 tumor slices from GFAT1-proficient and -deficient tumors from the subcutaneous and the orthotopic models. Blinded scoring using a 0–3 scale (*Figure 4G*) revealed more HA in the subcutaneous than orthotopic setting (*Figure 4H*). Tumor genotype did not influence HABP levels in the subcutaneous model, whereas there was a marked reduction in HABP staining in the orthotopic GFAT1 knockout tumors (*Figure 4H*). First, these data demonstrate that HA is available to PDA tumors in these models. Further, they suggest that the difference in growth of GFAT1 knockout tumors in the orthotopic model compared to subcutaneous, as observed in *Figure 1E–G*, may result from less HA availability.

## HA rescues GFAT1 null PDA via the GlcNAc salvage pathway

The GlcNAc salvage pathway bypasses GFAT1 by catalyzing the phosphorylation of GlcNAc to GlcNAc-6-phsophate, in a reaction mediated by *N*-acetyl-glucosamine kinase (NAGK). This

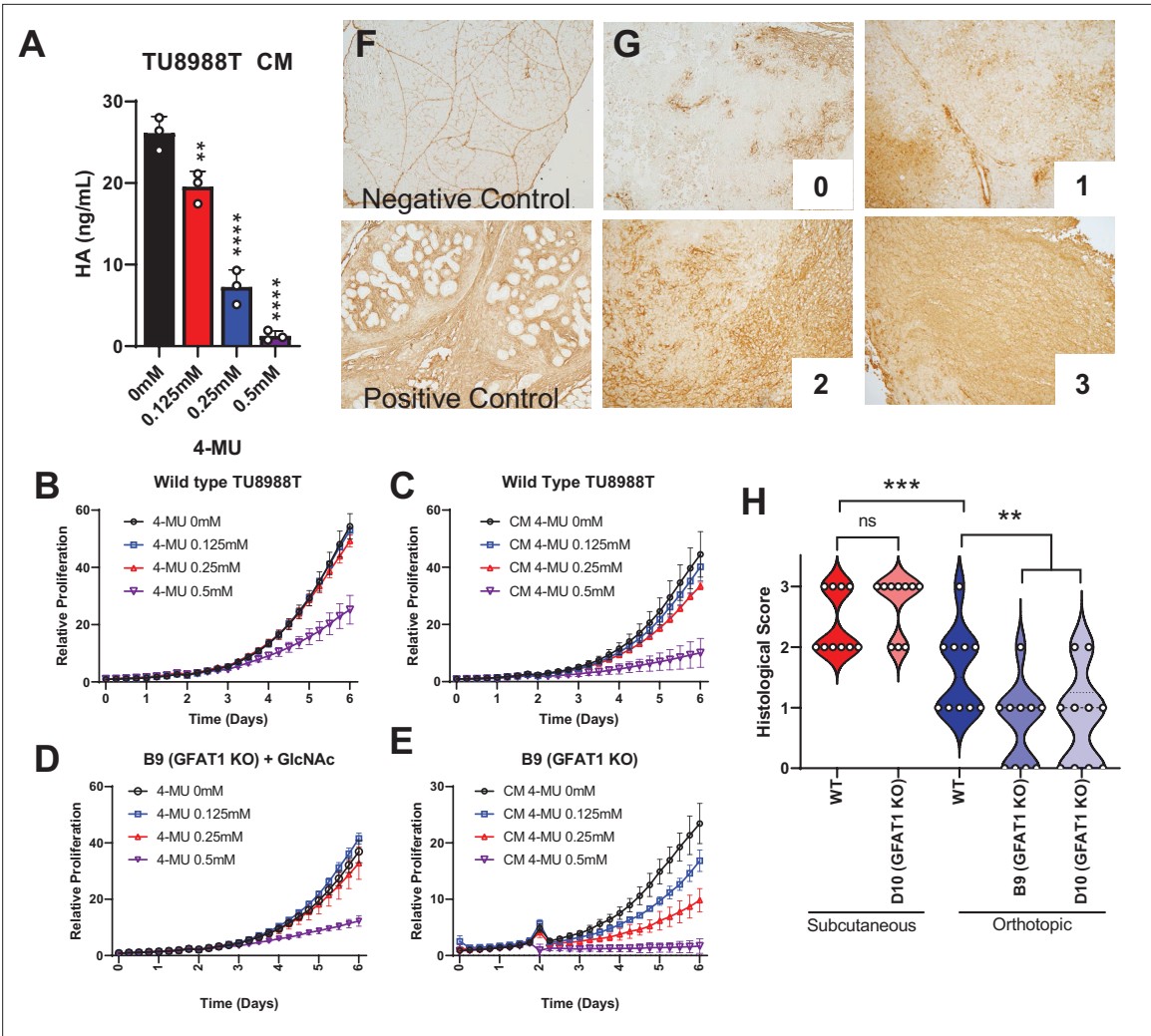

**Figure 4.** Hyaluronic acid in conditioned media (CM) rescues GFAT1 knockout. (**A**) Quantification of hyaluronic acid (HA) in CM from wildtype TU8988T cells treated with varying doses of 4-methylumbelliferone (4-MU) (n = 3). (**B–E**) Proliferation time course of (**B,C**) wildtype (WT) TU8988T and (**D,E**) GFAT1 knockout TU8988T grown (**B,D**) directly in varying concentrations of 4-MU or (**C,E**) in CM from WT TU8988T cells exposed to 4-MU during media conditioning. GFAT1 knockout cells in (**D**) were propagated in *N*-acetyl-glucosamine (GlcNAc) to maintain viability (n = 3 for all cell lines and conditions). (**F**) HA binding protein (HABP) staining of normal murine pancreas (negative control) and a murine pancreatic tumor (positive control). (**G**) Representative images for HABP staining classification used in **H**. (**H**) 10 representative slides from WT and GFAT1 knockout subcutaneous and orthotopic tumors (*Figure 1F and G*) were stained and blindly scored using the classification metric in **G** (n = 10). Error bars represent mean ± SD. *p < 0.05; **p < 0.01; ***p < 0.001; ****p < 0.0001.

GlcNAc-6-phosphate is subsequently converted into UDP-GlcNAc (*Figure 1A*). Therefore, we hypothesized that the carbohydrate polymer HA, which is 50% GlcNAc, fuels the HBP via the GlcNAc salvage pathway through NAGK. To test this, we employed the same CRISPR/Cas9 strategy to target NAGK (*Figure 5A*). Knockout of NAGK in parental TU8988T and MiaPaCa2 cell lines had no impact on colony formation, while reducing the colony-forming capacity of HPAC cells (*Figure 5B and C*). These results correlated with the elevated expression of NAGK in wildtype HPAC cells (*Figure 5D*). Of note, NAGK knockout did not result in up-regulation of GFAT1 (*Figure 5A*), which could have suggested a compensatory metabolic rewiring.

Next, we targeted NAGK in our GFAT1 knockout clones. GFAT1/NAGK double knockout cells were generated in media containing *N*-acetyl-galactosamine (GalNAc), an isomer of GlcNAc. Supplementation with GalNAc enables bypass of both the de novo HBP and the GlcNAc salvage pathway, by way of the Leloir pathway (*Termini et al., 2017*), to support UDP-GalNAc and ultimately UDP-GlcNAc

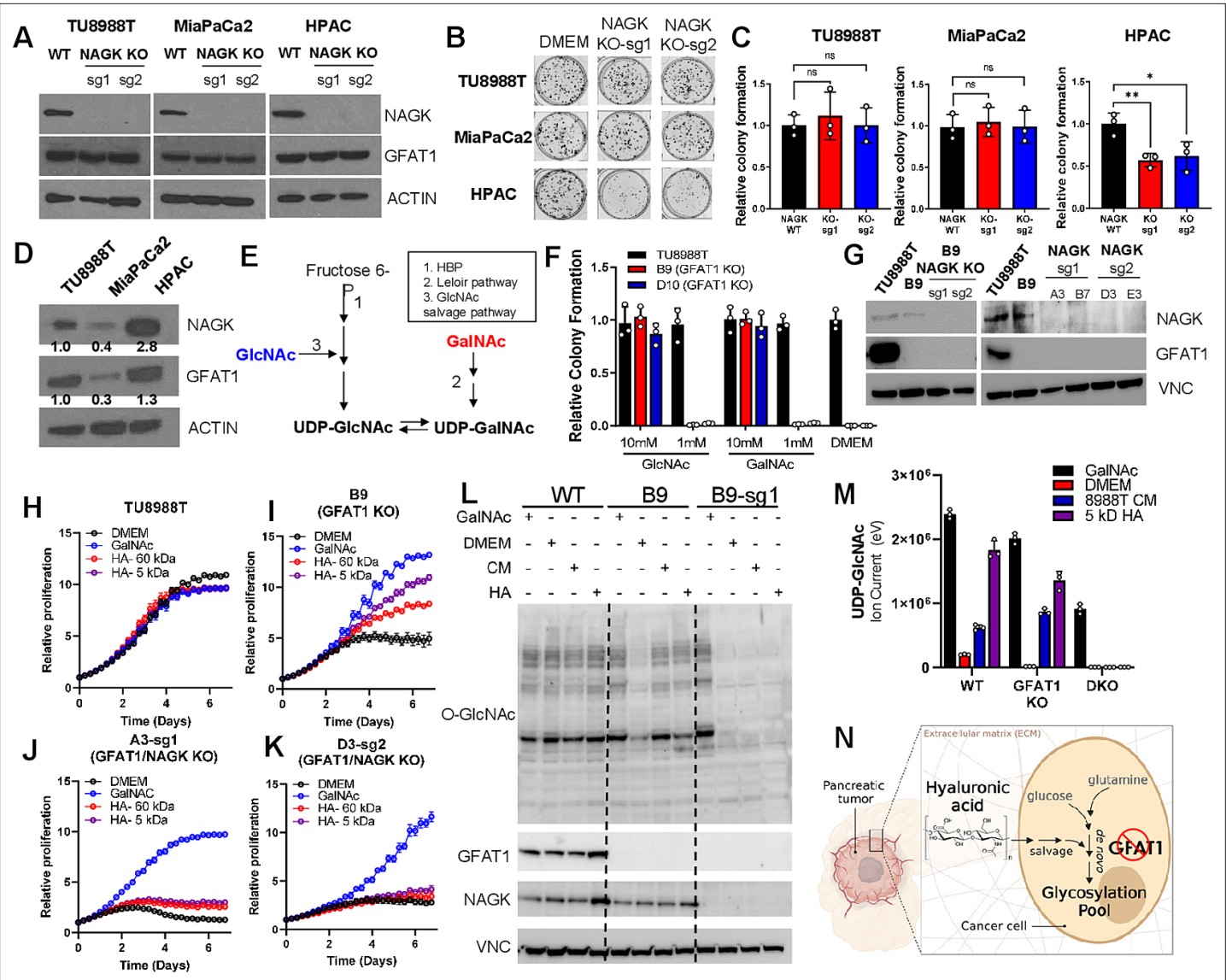

**Figure 5.** Hyaluronic acid-derived *N*-acetyl-glucosamine (GlcNAc) rescues GFAT1 loss via the GlcNAc salvage pathway. (**A**) Western blot of NAGK, GFAT1, and ACTIN loading control from TU8988T, MiaPaCa2, and HPAC parental (wildtype, WT) and NAGK knockout (KO) populations. NAGK was knocked out using two independent sgRNAs (sg1, sg2). (**B**) Representative wells from a colony-forming assay for parental and NAGK knockout lines. (**C**) Quantitation of colony-forming assay data in **B** (n = 3). (**D**) Western blot for NAGK, GFAT1, and loading control ACTIN in parental pancreatic ductal adenocarcinoma (PDA) cell lines. Band density was quantitated, normalized to control, and presented below the blot. (**E**) Schematic overview of the Leloir pathway of galactose catabolism integrated with the hexosamine biosynthetic pathway (HBP) and GlcNAc salvage pathway. (**F**) Quantitated data from colony formation assays in parental and GFAT1 knockout clonal TU8988T cell lines in base media (DMEM), positive control GlcNAc, and *N*-acetyl-galactosamine (GalNAc) (n = 3) (**G**) Western blot for GFAT1, NAGK, and loading control VINCULIN (VNC) in parental TU8988T and HPAC, GFAT1 knockout clones, and GFAT/NAGK double targeted lines. (**H–K**) Proliferation time course of (**H,I**) parental TU8988T and GFAT1 knockout line B9 in base media, GalNAc positive control, 60 kDa HA, or 5 kDa HA; (**J,K**) GFAT1/NAGK double targeted clones in base media, GalNAc positive control, 60 kDa HA, or 5 kDa hyaluronic acid (HA) (n = 3). (**L**) Western blot for proteome *O*-GlcNAcylation (*O*-GlcNAc), GFAT1, NAGK, and VCN in parental (WT), GFAT1 knockout (**B9**), and GFAT1/NAGK double knockout (B9–sg1) TU8988T cells treated with 10 mM GalNAc, DMEM, CM, or o-HA. (**M**) Liquid chromatography-coupled tandem mass spectrometry (LC-MS/MS) analysis of UDP-GlcNAc from the samples in **L** (n = 3). (**N**) Schematic overview of the HA metabolism through the GlcNAc salvage pathway to fuel glycosylation in GFAT1 knockout PDA. Error bars represent mean ± SD. n.s., non-significant; *p < 0.05; **p < 0.01.

The online version of this article includes the following figure supplement(s) for figure 5:

**Figure supplement 1.** Additional characterization of hyaluronic acid (HA) rescue in GFAT1/NAGK double knockout cell lines.

biogenesis (*Figure 5E*). In this way, we were again able to select viable lines while avoiding the selection of those with unpredictable metabolic adaptations.

The GalNAc dose response for GFAT1 knockout clones was comparable to that of GlcNAc (*Figure 2H*), demonstrating that they are indeed viable in GalNAc (*Figure 5F*, *Figure 3—figure supplement 2H*). Although NAGK expression was efficiently knocked down in our pooled populations (*Figure 5G*), we again selected for clones in the TU8988T cell line in order to minimize the effect of NAGK-proficient clones persisting in the bulk population. From among these, we selected four GFAT1/NAGK clones and tracked their proliferation upon rescue with varying sizes of HA, GalNAc, or PDA CM. These were compared relative to wildtype TU8988T cells and the GFAT1 knockout line. In stark contrast to the GFAT1 knockout line, LMW HA, o-HA, and PDA CM were unable to rescue GFAT1/NAGK double knockout lines (*Figure 5H–K*, *Figure 5—figure supplement 1A-F*). Similar results were obtained in the MiaPaCa2 cell line (*Figure 5—figure supplement 1G-J*).

Finally, we used LC/MS-MS to assess HBP metabolite levels and western blotting to follow proteome *O*-GlcNAcylation in the GFAT1 and GFAT1/NAGK double knockout cells. HA or PDA CM rescue of GFAT1 knockout cells restores UDP-GlcNAc pools and proteome *O*-GlcNAcylation, and this was blocked by knocking out NAGK (*Figure 5L and M* and *Figure 5—figure supplement 1K-N*). These results illustrate that HA rescue requires NAGK and the GlcNAc salvage pathway, consistent with the idea that HA-derived GlcNAc fuels UDP-GlcNAc biosynthesis upon GFAT1 knockout. Altogether, our data implicate HA as a novel nutrient for PDA, where HA regulates PDA metabolism by refueling the HBP via the GlcNAc salvage pathway. This supports PDA survival and proliferation (*Figure 5N*).

## Discussion

The HBP is activated in a Kras-dependent manner in PDA via transcriptional regulation of *Gfpt1* (*Ying et al., 2012*), and it is similarly elevated in numerous cancers to provide a diverse set of functions, including the regulation of proliferation, survival, angiogenesis, and metastasis (*Akella et al., 2019*). As such, we and others have proposed that the HBP may provide a selective vulnerability for cancer therapy, with GFAT1 as an attractive therapeutic target (*Ying et al., 2012*; *Walter et al., 2020*; *Guillaumond et al., 2013*; *Lucena et al., 2016*). Of note, GFAT2 is a homolog of GFAT1, and it too has been implicated as a drug target with context and tissue-specific functions (*Kim et al., 2020*; *Zhang et al., 2018*). We did not pursue GFAT2 in this study because it was neither regulated by mutant KRAS nor basally expressed in the models employed herein.

Several pan glutamine-deamidase inhibitors (e.g. azaserine and 6-diazo-5-oxo-L-norleucine), which potently suppress GFAT activity, have demonstrated anti-tumor activity in vitro and in vivo in PDA and other cancers (*Leone et al., 2019*; *Sharma et al., 2020*). However, because these drugs are not specific to the HBP, it has not been clear what impact GFAT-specific inhibition had on these phenotypes. As such, we took a genetic approach to knock out GFAT1 to elucidate the role of the HBP in PDA. In the PDA models tested, we found that GFAT1 knockout was not compatible with PDA cell proliferation in vitro, unless the media were supplemented with GlcNAc or GalNAc (*Figures 1C, D , and 5F*). However, these same cells readily formed tumors in vivo in subcutaneous and orthotopic models (*Figure 1E–G*).

The stark discrepancy in phenotypes led us to hypothesize that the TME was providing the means to bypass GFAT1. Indeed, we found that denatured CM from CAFs and wildtype PDA cells were able to rescue viability in GFAT1 knockout PDA cells, implicating a molecule(s) without tertiary structure (*Figure 2*). By examining several GlcNAc-containing candidates, we discovered a previously unknown role of HA as a nutrient source for PDA (*Figures 3 and 4*). Of note, despite identifying that wildtype PDA cells can produce HA, it is not our intention to indicate that HA in PDA tumors was deposited by the cancer cells. For example, HA is abundantly deposited in GFAT1 knockout subcutaneous PDA tumors (*Figure 4H*). Rather, we found that wildtype PDA CM could rescue the proliferation of GFAT1 knockout PDA cells in vitro, and we utilized this as a tool to study the process. The constellation of cell types in the pancreatic TME that produce/deposit/process HA is an area of active investigation. Either way, in sum we report that HA can refill the HBP via the GlcNAc salvage pathway to support PDA survival and proliferation (*Figure 5N*).

HA is traditionally regarded as a structural component in physiology (*Toole, 2004*). In addition to this role, a wealth of studies have ascribed other functions to HA. For example, HA can activate cell-cell contact-mediated signal transduction through CD44 and/or receptor for HA-mediated motility

(*Misra et al., 2015*). The signaling activity/function of HA depends on its MW (*Toole, 2004*; *Vigetti et al., 2014*). Similarly, a recent study illustrated that breakdown of the HA matrix with hyaluronidase enabled the interaction between growth factors and growth factor receptors (*Sullivan et al., 2018*). This promoted glucose metabolism, cellular proliferation, and migration. The role of HA in GFAT1 knockout PDA cells described herein is likely independent of its structural and signaling functions, given that we observe considerably greater rescue with o-HA (*Figure 3H, I*), a form of HA that is not traditionally considered for these purposes. Together with the increased rescue by o-HA, several additional experiments also suggested that intracellular catabolism of HMW HA impedes its use as a fuel. Namely, while HMW HA-FITC is readily captured by PDA cells (*Figure 3F*), HAase catabolism of HA potentiates rescue of GFAT1 (*Figure 3G*).

Our study introduces a novel role to HA as a fuel for PDA tumor growth (*Figure 3H, I*), further highlighting the significance and biological complexity of this predominant glycosaminoglycan. Additionally, our study suggests that NAGK, through which HA-mediated GlcNAc presumably refuels the HBP in vivo, may be an attractive therapeutic target for PDA. Indeed, a recent study demonstrated that NAGK knockout in PDA impairs tumor growth in vivo, while only exhibiting a modest impact on cellular proliferation in vitro (*Campbell et al., 2021*). These results are consistent with our observations that the GlcNAc salvage pathway is used to fill UDP-GlcNAc pools with HA-derived GlcNAc (*Figure 5N*). Our study also contributes to a growing body of data illuminating unexpected nutrient sources in the TME that support cancer metabolism (*Commisso et al., 2013*; *Kamphorst et al., 2015*; *Davidson et al., 2017*; *Olivares et al., 2017*; *Sousa et al., 2016*; *Dalin et al., 2019*; *Halbrook et al., 2019*; *Zhu et al., 2020*; *Jayashankar and Edinger, 2020*), and this raises the possibility that other glycosaminoglycans may be similarly scavenged.

Due to its extremely hydrophilic nature, HA retains water and acts as a cushioning agent in tissue homeostasis and biomechanical integrity (*Toole, 2004*). In PDA, HA is a predominant component of the TME, and its water-retaining property is one of the main drivers of the supraphysiological intratumoral pressure (*DuFort and DelGiorno, 2016*). This pressure can exceed 10-fold that observed in the normal pancreas, and, as a result, tumor vasculature collapses (*Provenzano et al., 2012*; *DuFort et al., 2016*; *Jacobetz et al., 2013*). The limited access to circulation impairs nutrient and oxygen delivery, and it has been proposed that this is a critical impediment to tumoral drug delivery (*Olive et al., 2009*). Indeed, in animal models, breakdown of the HA matrix with a therapeutic hyaluronidase (PEGPH20) reduces intratumoral pressure, restores circulation, which facilitates drug delivery, and thereby improves response to chemotherapy (*Provenzano et al., 2012*; *Jacobetz et al., 2013*). Based on these promising observations, PEGPH20 was tested in clinical trials alongside standard of care chemotherapy. Despite the successes in the preclinical models, PEGPH20 did not extend PDA patient survival (*Van Cutsem et al., 2020*).

The discrepancy between the clinical response to PEGPH20 and the preclinical data remains an active area of investigation and may concern the myriad additional roles of HA. For example, the HA matrix may be necessary to restrain tumor dissemination, as was shown for CAF depletion studies in PDA (*Helms et al., 2020*; *Lee et al., 2014*; *Özdemir et al., 2015*; *Rhim et al., 2014*). Thus, the benefits afforded by enhanced drug penetration facilitated by PEGPH20 may be negated by this side effect. Along these lines, HA degradation may also enhance tumor metabolism and growth. This could occur through growth factor signaling-dependent (*Sullivan et al., 2018*) as well as signaling-independent pathways, like the GlcNAc salvage pathway described herein. In contrast, reduction in the HA content of tumors also facilitates T cell invasion (*Sharma et al., 2020*), which may complement immunotherapy approaches, a concept that would be hindered by immunosuppressive chemotherapies. Given the conflicting roles of HA in tumor restraint and tumor growth, considerable work remains to be done to determine the most effective way to exploit this feature of pancreatic cancer.

# Materials and methods
## Cell culture
MiaPaCa2 (ATCC Cat# CRM-CRL-1420, RRID:CVCL_0428) and HPAC (ATCC Cat# CRL-2119, RRID:CVCL_3517) were obtained from ATCC. TU8988T (DSMZ Cat# ACC-162, RRID:CVCL_1847) was obtained from DSMZ. Patient-derived CAFs (*Hwang et al., 2008*) were a generous gift from Rosa Hwang, and FL5.12 cells were a generous gift from Dr Aimee Edinger. All cells were routinely checked

for mycoplasma contamination with MycoAlert PLUS (Lonza) and validated for authenticity annually by STR profiling. Cells were maintained in standard high glucose DMEM without pyruvate (Gibco) supplemented with 10% fetal bovine serum (FBS; Corning). GFAT1 null PDA were cultured in standard media supplemented with 10 mM GlcNAc (Sigma). GFAT1 null NAGK knockout PDA were cultured in standard media supplemented with 10 mM GalNAc (Sigma). Low nutrient media was made with DMEM without glucose, glutamine, and pyruvate (Gibco). Glucose, glutamine, and FBS were added to the final concentration of 1.25 mM, 0.2 mM, and 1%, respectively. FL5.12 cells were maintained in RPMI 1640 (Gibco) supplemented with 10% FBS, 10 mM HEPES (Sigma), 55 µM β-mercaptoethanol (Sigma), antibiotics, 2 mM glutamine, and 500 pg/mL recombinant murine IL-3 (Peprotech 213–13).

### Generation of CRISPR/Cas9 knockout clones

GFAT1 and NAGK knockout PDA cell lines were generated using CRISPR/Cas9 method described previously (*Zhu et al., 2020*). Overlapping oligonucleotides (Feng Zhang lab human GeCKOv2 CRISPR knockout pooled library; Addgene #1000000048) were annealed to generate sgRNA targeting GFAT1 or NAGK. sgRNA was cloned directly into the overhangs of PX459 V2.0 vector (Feng Zhang lab; Addgene plasmid #62988) that was digested with BbsI. The resulting CRISPR/Cas9 plasmid was transformed into chemically competent Stbl3 cells, miniprepped for plasmid DNA, and sequence-verified. sgRNA oligonucleotide pairs for GFAT1 (11) and NAGK are as follows: GFAT1 (sg1 Fwd 5'-CACCgCTTCAGAGACTGGAGTACAG-3'; sg1 Rev 5'-AAACCTGTACTCCAGTCTCTGAAGc-3') and NAGK (sg1 Fwd 5'-CACCgTAGGGGAGGCACACGATCCG; sg1 Rev 5'-AAACCGGATCGTGTGC CTCCCCTAc-3'; sg2 Fwd 5'-CACCgGCCTAGGGCCTATCTCTGAG-3'; sg2 Rev 5'-AAACCTCAGAGA TAGGCCCTAGGCc-3'). Human PDA were transiently transfected using Lipofectamine 3000 according to the manufacturer's instructions. Cells were selected with puromycin in the presence of GlcNAc (GFAT1 knockout bulk population) or GalNAc (GFAT1 NAGK double knockout bulk population). To select clones, polyclonal pools were seeded into 96-well plates at a density of one cell per well. Individual clones were expanded and verified via western blot and Sanger sequencing. TU8988T clone B9 has a 10 base pair (bp) and a 1 bp deletion in GFAT1; TU8988T clone D10 has two different 1 bp deletions in GFAT1; MiaPaCa2 clone M12 has two different 1 bp deletions in GFAT1; HPAC clone H1 has a 187 bp deletion in GFAT1; HPAC clone H7 has a 187 bp deletion in GFAT1.

### Conditioned media

CM was generated by culturing cells in 15 cm² plates (25 mL growth media/plate) for 72 hr at 37°C, 5% CO₂, so that they reached ~90% confluence. The media were then filtered through a 0.45 µm polyethersulfone membrane (VWR). Boiled CM was warmed to 100°C for 15 min. Freeze-thaw CM were initially stored at –80°C and were thawed in a 37°C water bath on the day of the experiment. As indicated, fresh growth media were added to the CM at the ratios indicated to avoid nutrient/metabolite exhaustion.

### Colony formation and proliferation assays

For colony formation assays, cells were plated in a six-well plate in biological triplicate at 500 cells/well in 2 mL of media and grown for 9–12 days. For proliferation assays, 5000 cells/well were plated. At endpoint, assays were fixed with 100% methanol for 10 min and stained with 0.5% crystal violet solution for 15 min. Relative colony formation was quantitated manually in a blinded fashion. Proliferation was quantified by removing the dye with 10% acetic acid and measuring OD595.

### CyQUANT viability assay

Cells were seeded in 96-well black wall, clear bottom plates at 1000 cells/well in 50 µL of media and incubated at 37°C, 5% CO₂ for indicated time points. At each time point, media was aspirated and plates were stored at –80°C. Proliferation was determined by CyQUANT (Invitrogen) according to the manufacturer's instructions. SpectraMax M3 Microplate reader (Molecular Devices) was used to measure fluorescence.

## IncuCyte S3: real-time, live-cell proliferation assay

1000 cells were seeded per well in a 96-well plate and incubated at 37°C, 5% $CO_2$ for cells to equilibrate. The next day, media were aspirated, washed once with PBS, and replaced with different media as indicated. Proliferation was measured on IncuCyte S3 using phase object confluence as a readout.

## HA-FITC uptake experiments

TU8988T cells (WT, B9-GFAT1KO, D10-GFAT1KO) and BxPC3 cells were seeded in six-well plates with DMEM + 10% FBS + GlcNAc. The next day, cells were rinsed with PBS and the media was switched to DMEM + 10% FBS. The following day, cells were seeded on a chamber slide and grown in DMEM + 10% FBS. MiaPaCa2 cells (WT, M12-GFAT1KO) were treated similarly and were seeded in six-well plates with DMEM + 10% FBS + GlcNAc. The next day, cells were rinsed with PBS and seeded on a chamber slide in DMEM + 10% FBS.

After 1 day in chamber slides, the media was removed, cells were rinsed with PBS, and cells were incubated with HA-FITC media for 1 hr at 37°C (or 1 min at 37°C as a negative control). After incubation, cells were rinsed and fixed with 10% formalin for 15 min at room temperature. Wells were rinsed three times with PBS + 0.3% BSA and cells were blocked for 1 hr at room temperature with PBS + 1% BSA. Cells were incubated overnight at 4°C with Mouse anti-panCK (Cytokeratin, DAKO M3515, [1:100]) in PBS + 1% BSA. The following day, cells were rinsed three times with PBS + 0.3% BSA and incubated 1 hr at room temperature with an anti-mouse Alexa Fluor secondary antibody (Invitrogen) in PBS + 1% BSA. Following incubation, cells were rinsed and mounted with ProLong Gold Antifade Mountant with DAPI (Invitrogen).

Olympus BX53F microscope, Olympus DP80 digital camera, and CellSens Standard software were used for imaging and six representative images were taken per cell line and condition. Uptake index was calculated using the Analyze Particles feature in ImageJ after automatic thresholding. The cell outlines and regions of interest were determined using panCK expression and HA-FITC particles were measured. HA-FITC particle area was plotted as a percent of total cell area.

## HA ELISA

An HA ELISA kit (Cat #DY3614, R&D Systems) was used to determine the concentration of HA in all the different CM samples as per manufacturer's instructions.

## Metabolite sample Preparation

Intracellular metabolite fractions were prepared from cells grown in six-well plates. The media was aspirated, and cells were incubated with cold (−80°C) 80% methanol (1 mL/well) on dry ice for 10 min. Then, the wells were scraped with a cell scraper and transferred to 1.5 mL tubes on dry ice. To measure GlcNAc concentration in different CM, 0.8 mL of ice-cold 100% methanol was added to 0.2 mL of CM, and the mixture was incubated on dry ice for 10 min.

After incubation of cell or media fractions on dry ice, the tubes were centrifuged at 15,000 rpm for 10 min at 4°C to pellet the insoluble material, and the supernatant was collected in a fresh 1.5 mL tube. Metabolite levels of intercellular fractions were normalized to the protein content of a parallel sample, and all samples were lyophilized on a SpeedVac, and re-suspended in a 50:50 mixture of methanol and water in HPLC vials for LC-MS/MS analysis.

## Liquid chromatography-coupled mass spectrometry

To detect UDP-GlcNAc, the Shimadzu NEXERA integrated UHPLC system with an LC30AD pump, SIL30AC autosampler, CTO30A column oven, CBM20A controller was coupled with the AB Sciex TripleTOF 5600 MS system with DuoSpray ion source. All calibrations and operations were under control of Analyst TF 1.7.1. Calibrations of TOF-MS and TOF-MS/MS were achieved through reference APCI source of SCEIX calibration solution. A high-throughput LC method of 8 min with flowrate of 0.4 mL/min with a Supelco Ascentis Express HILIC (75 mm × 3.0 mm, 2.7 µm). Solvent A was made of 20 mM ammonium acetate of 95% water and 5% acetonitrile at pH 9.0. Solvent B was 95% acetonitrile and 5% water. LC gradient 0.0–0.5 min 90% B, 3 min 50% B, 4.10 min 1% B, 5.5 min 1% B, 5.6 min 90% B, 6.5 min 90% B, 8 min stopping. Key parameters on the MS were the CE and CE spread of −35 ev, 15 ev. Data were compared to a reference standard. Data processing was performed by Sciex PeakView, MasterView, LibraryView and MQ software tools, and ChemSpider database.

To measure GlcNAc concentration in the various CM, we utilized an Agilent Technologies Triple Quad 6470 LC/MS system consisting of 1290 Infinity II LC Flexible Pump (Quaternary Pump), 1290 Infinity II Multisampler, 1290 Infinity II Multicolumn Thermostat with six port valve and 6470 triple quad mass spectrometer. Agilent Masshunter Workstation Software LC/MS Data Acquisition for 6400 Series Triple Quadrupole MS with Version B.08.02 is used for compound optimization and sample data acquisition.

A GlcNAc standard was used to establish parameters, against which CM were analyzed. For LC, an Agilent ZORBAX RRHD Extend-C18, 2.1 × 150 mm, 1.8 μm and ZORBAX Extend Fast Guards for UHPLC were used in the separation. LC gradient profile is: at 0.25 mL/min, 0–2.5 min, 100% A; 7.5 min, 80% A and 20% C; 13 min 55% A and 45% C; 20 min, 1% A and 99% C; 24 min, 1% A and 99% C; 24.05 min, 1% A and 99% D; 27 min, 1% A and 99% D; at 0.8 mL/min, 27.5–31.35 min, 1% A and 99% D; at 0.6 mL/min, 31.50 min, 1% A and 99% D; at 0.4 mL/min, 32.25–39.9 min, 100% A; at 0.25 mL/min, 40 min, 100% A. Column temperature is kept at 35°C, samples are at 4°C, injection volume is 2 μL. Solvent A is 97% water and 3% methanol 15 mM acetic acid and 10 mM tributylamine at pH of 5. Solvent C is 15 mM acetic acid and 10 mM tributylamine in methanol. Washing Solvent D is acetonitrile. LC system seal washing solvent 90% water and 10% isopropanol, needle wash solvent 75% methanol, 25% water.

6470 Triple Quad MS is calibrated with ESI-L Low concentration Tuning mix. Source parameters: gas temperature 150°C, gas flow 10 L/min, nebulizer 45 psi, sheath gas temperature 325°C, sheath gas flow 12 L/min, capillary –2000 V, delta EMV –200 V. Dynamic MRM scan type is used with 0.07 min peak width, acquisition time is 24 min. Delta retention time of plus and minus 1 min, fragmentor of 40 eV and cell accelerator of 5 eV are incorporated in the method.

## Xenograft studies

Animal experiments were conducted in accordance with the Office of Laboratory Animal Welfare and approved by the Institutional Animal Care and Use Committees of the University of Michigan. NOD-SCID gamma (NSG) mice (Jackson Laboratory), 6–10 weeks of age of both sexes, were maintained in the facilities of the Unit for Laboratory Animal Medicine (ULAM) under specific pathogen-free conditions. Protocol#: PRO00008877.

Wildtype TU8988T and two verified GFAT1 null clones (B9 and D10) were trypsinized and suspended at 1:1 ratio of DMEM (Gibco, 11965–092) cell suspension to Matrigel (Corning, 354234); 150–200 μL were used per injection. Orthotopic tumors were established by injecting 0.5 × 10⁶ cells in 50 μL of 1:1 DMEM to Matrigel mixture. The experiment lasted 5 weeks. For subcutaneous xenograft studies with the pooled populations or validated clones, tumors were established with 5 × 10⁶ cells in 200 μL of 1:1 DMEM to Matrigel mixture.

Tumor size was measured with digital calipers two times per week. Tumor volume (V) was calculated as $V = 1/2(\text{length} \times \text{width}^2)$. At endpoint, final tumor volume and mass were measured prior to processing. Tissue was snap-frozen in liquid nitrogen, then stored at −80°C.

## Western blot analysis

After SDS-PAGE, proteins were transferred to PVDF membrane, blocked with 5% milk, and incubated with primary antibody overnight at 4°C. The membranes were washed with TBST, incubated with the appropriate horseradish peroxidase-conjugated secondary antibody for 1 hr and visualized on Bio-Rad imager with enhanced chemiluminescence detection system or exposed on radiographic film.

## Immunohistochemistry on subcutaneous and orthotopic tumors

Subcutaneous and orthotopic tumors were fixed in Z-fix overnight, paraffin embedded, and sectioned onto slides. Sections were deparaffinized in xylene, rehydrated, and blocked with 2.5% BSA prior to incubation with biotinylated HABP antibody (Calbiochem #385911, [1:200]) overnight at 4°C. Vector Laboratories Vectastain Elite ABC-HRP Kit (PK-6100) and Vector Laboratories DAB Substrate Kit (SK-4100) were used for peroxidase detection of HABP signal.

## Histological scoring

HABP-stained tumors, normal pancreas tissue (negative control), and transformed pancreas tissue from KC mice (positive control) were processed and stained for HABP as described above. Ten

representative 20× images from each group were scored blinded based on HABP staining. Staining was scored on the following scale: 0, no staining; 1, minimal staining; 2, moderate to strong staining in at least 20% of cells; 3, strong staining in at least 50% of cells.

## Antibodies

The following antibodies were used in this study: VINCULIN (Cell Signaling 13901), ACTIN (Santa Cruz sc-47778), GAPDH (Cell Signaling 5174), GFAT1 (Abcam 125069), NAGK (Atlas Antibodies HPA035207), O-GlcNAc (Abcam 2735), panCK (Cytokeratin, DAKO M3515), biotinylated HABP (Calbiochem 385911), secondary anti-mouse-HRP (Cell Signaling 7076), and secondary anti-rabbit-HRP (Cell Signaling 7074).

## Detection and quantification of macropinocytosis

The macropinocytosis index was measured as previously described (*Commisso et al., 2014*). In brief, cells were seeded on the coverslips in 24-well plate for 24 hr and serum-starved for 18 hr. Cells were incubated with 1 mg/mL HMW TMR-dextran (Fina Biosolutions) in serum-free medium for 30 min at 37°C. Cells were then washed five times with cold DPBS and fixed in 4% polyformaldehyde for 15 min. The coverslips were mounted onto slides using DAKO Mounting Medium (DAKO) in which nuclei were stained with DAPI. At least six images were captured for each sample using an Olympus FV1000 confocal microscope and analyzed using the particle analysis feature in ImageJ (NIH). The micropino-cytosis index for each field was calculated as follows: Macropinocytosis index = (total particle area/total cell area) × 100%.

## HA, hyaluronidase, and heparin

Heparin was obtained from Sigma (H3393). Oligo HA (5 kDa) was obtained from Lifecore Biomedical. Two different LMW HA were used in this study: 78 kDa HA (Pure Health Solutions) and 60 kDa HA (Lifecore Biomedical). To make 10 mM oligo- or LMW HA media, HA was added slowly into high glucose DMEM without pyruvate, stirred for 2 hr at room temperature, and filtered through 0.20 µm polyethersulfone membrane. FBS was added to a final concentration of 10%.

Hyaluronidase (Sigma H3506) treatment was performed as follows: 10 mM LMW HA media and control media (DMEM +10% FBS) were incubated with hyaluronidase, according to manufacturer's instructions, overnight in a 37°C water bath. The next day, media were boiled for 15 min to denature hyaluronidase. The resulting media were mixed 1:1 with fresh growth media to avoid effects of nutrient/metabolite exhaustion.

## Preparation of necrotic FL5.12 cells

Necrotic FL5.12 cells were prepared as described previously (*Kim et al., 2018*). Cells were washed three times with PBS, cultured in the FL5.12 media without IL-3 (100 million cells/mL) for 72 hr. The necrotic cells were spun down at 13,000 rpm for 10 min at 4°C, and the pellets were stored at –80°C until use.

## Statistical analysis

Statistics were performed using GraphPad Prism 8. Groups of two were analyzed with two-tailed Student's t-test. Groups of more than two were analyzed with one-way ANOVA Tukey post hoc test. All error bars represent mean with standard deviation. A p value of less than 0.05 was considered statistically significant. All group numbers and explanation of significant values are presented within the figure legends.

## Acknowledgements

The authors would like to thank Dr Sunil Hingorani for helpful suggestions and Brandon Chen for preparation of the scheme in *Figure 5*. This work was funded by T32AI007413 and F31CA243344 (PK); K99/R00CA241357 and P30DK034933 (CJH); T32AI007413, F31CA24745701, and F99/K00CA264414 (SAK); T32HD007505 (MR); R01CA237466, R01CA252037, and R21CA212958 (KRK), StandUp2Cancer (KRK), Thompson Family Foundation (KRK), Geoffrey Beene Cancer Research Center at MSKCC and the STARR Cancer Consortium, as well as the MSKCC NIH/NCI Cancer Center Support Core Grant P30CA008748; a Pancreatic Cancer Action Network/AACR Pathway to Leadership award

(13-70-25-LYSS), Junior Scholar Award from The V Foundation for Cancer Research (V2016-009), Kimmel Scholar Award from the Sidney Kimmel Foundation for Cancer Research (SKF-16–005), a 2017 AACR NextGen Grant for Transformative Cancer Research (17-20-01-LYSS), the Cancer Center support grant (P30 CA046592); and NIH grants R37CA237421, R01CA248160, R01CA244931 (CAL). Metabolomics studies were supported by NIH grant DK097153, the Charles Woodson Research Fund, and the UM Pediatric Brain Tumor Initiative.

## Additional information

### Competing interests

Costas A Lyssiotis: has received consulting fees from Astellas Pharmaceuticals and Odyssey Therapeutics and is an inventor on patents pertaining to Kras regulated metabolic pathways, redox control pathways in pancreatic cancer, and targeting the GOT1-pathway as a therapeutic approach (US Patent No: 2015126580-A1, 05/07/2015; US Patent No: 20190136238, 05/09/2019; International Patent No: WO2013177426-A2, 04/23/2015). The other authors declare that no competing interests exist.

### Funding

| Funder | Grant reference number | Author |
|---|---|---|
| National Cancer Institute | Cancer Biology Training Grant,T32AI007413 | Peter K Kim Samuel A Kerk |
| National Cancer Institute | Predoctoral Fellowship F31CA243344 | Peter K Kim |
| National Cancer Institute | Pathway to Independence Award K99CA241357 | Christopher J Halbrook |
| National Institute of Diabetes and Digestive and Kidney Diseases | Postdoctoral Support P30DK034933 | Christopher J Halbrook |
| National Cancer Institute | F31CA24745701 | Samuel A Kerk |
| National Cancer Institute | R01CA237466 | Kayvan R Keshari |
| National Cancer Institute | R01CA252037 | Kayvan R Keshari |
| National Cancer Institute | R21CA212958 | Kayvan R Keshari |
| Stand Up To Cancer | Research Grant | Kayvan R Keshari |
| Thompson Family Foundation | Research Grant | Kayvan R Keshari |
| STARR Cancer Consortium | Research Grant | Kayvan R Keshari |
| National Cancer Institute | Cancer Center Support Grant P30CA008748 | Kayvan R Keshari |
| American Association for Cancer Research | Pathway to Leadership award 13-70-25-LYSS | Costas A Lyssiotis |
| V Foundation for Cancer Research | Junior Scholar Award V2016-009 | Costas A Lyssiotis |
| Sidney Kimmel Foundation | Kimmel Scholar Award SKF-16-005 | Costas A Lyssiotis |
| American Association for Cancer Research | NextGen Grant for Transformative Cancer Research 17-20-01-LYSS | Costas A Lyssiotis |
| National Cancer Institute | Cancer Center Support Grant P30 CA046592 | Costas A Lyssiotis |
| National Cancer Institute | R37CA237421 | Costas A Lyssiotis |
| National Cancer Institute | R01CA248160 | Costas A Lyssiotis |

| Funder | Grant reference number | Author |
|---|---|---|
| National Cancer Institute | R01CA244931 | Costas A Lyssiotis |
| National Institutes of Health | U24DK097153 | Costas A Lyssiotis |
| Charles Woodson Research Fund | Research Support | Costas A Lyssiotis |
| UM Pediatric Brain Tumor Initiative | Research Support | Costas A Lyssiotis |
| National Cancer Institute | F99/K00CA264414 | Samuel A Kerk |
| National Institute of Child Health and Human Development | T32HD007505 | Megan Radyk |

The funders had no role in study design, data collection and interpretation, or the decision to submit the work for publication.

## Author contributions
Peter K Kim, Conceptualization, Data curation, Formal analysis, Investigation, Methodology, Validation, Writing – original draft, Writing – review and editing; Christopher J Halbrook, Conceptualization, Data curation, Formal analysis, Investigation, Writing – review and editing; Samuel A Kerk, Data curation, Formal analysis, Conceptualization, Investigation, Methodology, Writing – review and editing; Megan Radyk, Data curation, Formal analysis, Methodology, Writing – review and editing; Stephanie Wisner, Peter Sajjakulnukit, Sean W Hou, Galloway Thurston, Abhinav Anand, Liang Yan, Samuel D Welling, Li Zhang, Data curation, Formal analysis; Daniel M Kremer, Ayush Trivedi, Formal analysis; Anthony Andren, Formal analysis, Investigation; Lucia Salamanca-Cardona, Data curation; Matthew R Pratt, Kayvan R Keshari, Haoqiang Ying, Conceptualization, Funding acquisition, Project administration, Supervision; Costas A Lyssiotis, Conceptualization, Funding acquisition, Project administration, Supervision, Writing – original draft, Writing – review and editing

## Author ORCIDs
Peter K Kim http://orcid.org/0000-0001-9382-7223
Matthew R Pratt http://orcid.org/0000-0003-3205-5615
Costas A Lyssiotis http://orcid.org/0000-0001-9309-6141

## Ethics
Animal experiments were conducted in accordance with the Office of Laboratory Animal Welfare and approved by the Institutional Animal Care and Use Committees of the University of Michigan. Protocol#: PRO00008877.

## Decision letter and Author response
Decision letter https://doi.org/10.7554/eLife.62645.sa1
Author response https://doi.org/10.7554/eLife.62645.sa2

# Additional files

## Supplementary files
• Source data 1. Raw western blot images.
• Transparent reporting form

## Data availability
All data generated or analysed during this study are included in the manuscript and supporting file; raw images have been provided for all western blots in the Source Data file.

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
