## [Editor Report]

In this manuscript, the authors report that a major component of the pancreatic cancer microenvironment, hyaluronic acid, provides an important source of metabolic intermediates required for pancreatic cancer growth during conditions of nutrient limitation. Overall, this work nicely combines multiple cell lines and genetic approaches to show that scavenging of N-acetyl-glucosamine from hyaluronic acid supports cancer cell growth when de novo synthesis of N-acetyl-glucosamine is limited. More broadly, this work highlights how reliance on certain metabolic enzymes can vary depending on whether cells are grown in vitro or in vivo and provides evidence that environmental nutrient scavenging can contribute to differential metabolic dependencies of cancer cells.

---

## [Decision Letter]

**Decision letter after peer review:**

Thank you for submitting your article "Hyaluronic Acid Fuels Pancreatic Cancer Growth" for consideration by *eLife*. Your article has been reviewed by 3 peer reviewers, and the evaluation has been overseen by a Reviewing Editor and Richard White as the Senior Editor. The reviewers have opted to remain anonymous.

The reviewers have discussed the reviews with one another and the Reviewing Editor has drafted this decision to help you prepare a revised submission.

Summary:

In this manuscript, the authors report that a major component of the pancreatic cancer microenvironment, hyaluronic acid, can support pancreatic cancer cell growth when de novo hexosamine biosynthesis is suppressed or during conditions of nutrient limitation. How cancer cells cope with metabolic challenges to sustain growth and survival in hostile environments is an area of great interest. Here, the authors address how cells might cope with limited de novo synthesis of the metabolic intermediate GlcNAc, which is required for protein glycosylation and is reported to be sensitive to fluctuations in nutrient availability. The authors propose the novel and interesting central hypothesis that pancreatic ductal adenocarcinoma (PDAC) cells overcome limited synthesis of the metabolic intermediate UDP-GlcNAc, presumably a consequence of nutrient limitation in the tumor microenvironment, by taking up hyaluronic acid (HA) from the extracellular space and scavenging GlcNAc from HA. That PDAC cells cannot proliferate in vitro without GFAT1 expression, but can form tumors in the orthotopic or subcutaneous setting, is a compelling finding and lends some support to the notion that components of an intact tumor microenvironment can override blockade of the HBP. However, the manuscript is without formal demonstration that PDAC cells indeed take up HA or HA fragments, or that HA-derived GlcNAc contributes to UDP-GlcNAc pools or to protein glycosylation. To strengthen their conclusions and illustrate the potential of HA scavenging for survival under nutrient depleted conditions, the authors should address the following points:

Essential Revisions:

1. HA fragments can signal via cell surface receptors to increase glucose uptake and profoundly impact glucose metabolism (perhaps including increased HBP activity mediated by GFAT2 in the GFAT1 KO setting). To rule out this and other alternative explanations for the data presented, it is important for the authors to provide direct evidence for HA uptake (for example, using commercially available fluorescent HA) and/or that GlcNAc from HA in fact contributes to GlcNAc/UDP-GlcNAc pools.

2. The authors should directly test their model that HA is a key component of conditioned medium or the tumor microenvironment that supports growth in conditions of low nutrients and/or GFAT1 deletion. Is HA present in conditioned medium? Does NAGK deletion prevent rescue of growth by conditioned medium? Alternatively, the authors could use pharmacological inhibitors of HA synthesis in vitro (using CAFs or PDAC cells) and compare the growth effect of HA-impaired conditioned medium relative to control conditioned medium to rescue growth in order to test whether HA is a necessary component of conditioned medium for growth rescue. Absent these experiments, the authors should test whether GFAT1/NAGK double KO cells can form tumors in vivo in order to determine whether, besides HA cleavage and GlcNAc salvage via NAGK, there are alternative nutrients/pathways that may support glycosylation and cellular proliferation.

3. A related point centers around the notable context-specific environmental rescue of GFAT1 deletion. For example, Figures 1F and G show a complete rescue of GFAT1 KO in the subcutaneous setting, but a significant impairment of GFAT1 KO tumor growth in the orthotopic setting. How do HA levels compare in these distinct environments? Similarly, the conditioned medium rescue data presented in Figure 2, particularly the increased rescue by PDAC cell conditioned medium compared to CAF conditioned medium, is difficult to interpret together with the in vivo results in Figure 1F and G which suggest that the mechanism underlying rescue of proliferation is independent of PDAC cells themselves. Do GFAT1-replete PDAC cells secrete HA, and if so, do the levels exceed HA in CAF conditioned medium? To address these questions, the authors could perform ELISA-based assays for HA present in conditioned medium and IHC for HA binding protein in vivo.

4. To illustrate the generalizability of their model, the authors should repeat key findings, including the ability of hyaluronic acid to rescue growth of cells cultured in low nutrients or with GFAT1 deletion, in at least 2 of the cell lines generated and used in the manuscript. Similarly, the results of HA supplementation in GFAT1/NAGK double KO cells (Figure 4H-M) lends support to the importance of hexosamine salvage for PDAC cell proliferation, but this is only shown in the 8988T cell line and should be reproduced in an additional PDAC cell line.

Title:

We recommend the authors change the title to "Hyaluronic Acid Fuels Pancreatic Cancer Cell Growth" to reflect the fact that experiments testing the role of hyaluronic acid supporting cancer cell growth was performed in vitro.

---

## [Author Response]

Summary:In this manuscript, the authors report that a major component of the pancreatic cancer microenvironment, hyaluronic acid, can support pancreatic cancer cell growth when de novo hexosamine biosynthesis is suppressed or during conditions of nutrient limitation. How cancer cells cope with metabolic challenges to sustain growth and survival in hostile environments is an area of great interest. Here, the authors address how cells might cope with limited de novo synthesis of the metabolic intermediate GlcNAc, which is required for protein glycosylation and is reported to be sensitive to fluctuations in nutrient availability. The authors propose the novel and interesting central hypothesis that pancreatic ductal adenocarcinoma (PDAC) cells overcome limited synthesis of the metabolic intermediate UDP-GlcNAc, presumably a consequence of nutrient limitation in the tumor microenvironment, by taking up hyaluronic acid (HA) from the extracellular space and scavenging GlcNAc from HA. That PDAC cells cannot proliferate in vitro without GFAT1 expression, but can form tumors in the orthotopic or subcutaneous setting, is a compelling finding and lends some support to the notion that components of an intact tumor microenvironment can override blockade of the HBP. However, the manuscript is without formal demonstration that PDAC cells indeed take up HA or HA fragments, or that HA-derived GlcNAc contributes to UDP-GlcNAc pools or to protein glycosylation. To strengthen their conclusions and illustrate the potential of HA scavenging for survival under nutrient depleted conditions, the authors should address the following points:

We would like to thank the referees for this well-articulated, concise summary and the series of helpful suggestions. We agree with the important limitations noted during the initial review regarding HA uptake and the filling of UDP-GlcNAc pools. In the detailed rebuttal that follows, we address these concerns in full, demonstrating that HA is captured and utilized by PDA cells, that this is incorporated into UDP-GlcNAc pools, and consequently this supports cellular proteome O-GlcNAcylation.

Essential revisions:1. HA fragments can signal via cell surface receptors to increase glucose uptake and profoundly impact glucose metabolism (perhaps including increased HBP activity mediated by GFAT2 in the GFAT1 KO setting). To rule out this and other alternative explanations for the data presented, it is important for the authors to provide direct evidence for HA uptake (for example, using commercially available fluorescent HA) and/or that GlcNAc from HA in fact contributes to GlcNAc/UDP-GlcNAc pools.

This is an excellent suggestion. First, we employed fluorescently-labeled (FITC) HA and demonstrate that this is readily engulfed in our wild type and the corresponding GFAT1 knockout PDA cell lines. Representative images and quantitative data are now presented in Figure 3F and Figure 3—figure supplement 1H of the revised manuscript. Second, we employed mass spectrometry to follow hexosamine biosynthetic pathway (HBP) metabolites in wild type, GFAT1 knockout, and GFAT1/NAGK double knockout cells treated with HA or PDA conditioned media (CM). As is illustrated in Figure 5M and Figure 5—figure supplement 1K, HA or CM rescue GlcNAc-P and UDP-GlcNAc pools in the GFAT1 knockout PDA cells. This activity is lost in the double knockout cells, indicating the role of the GlcNAc salvage pathway in HA metabolism. Identical trends are observed in the wildtype, KO, and DKO cells when western blotting for proteome O-GlcNAcylation (Figure 5L and Figure 5—figure supplement 1N) and when tracking proliferation (Figure 5H-K, Figure 5—figure supplement 1A-J).

2. The authors should directly test their model that HA is a key component of conditioned medium or the tumor microenvironment that supports growth in conditions of low nutrients and/or GFAT1 deletion. Is HA present in conditioned medium? Does NAGK deletion prevent rescue of growth by conditioned medium? Alternatively, the authors could use pharmacological inhibitors of HA synthesis in vitro (using CAFs or PDAC cells) and compare the growth effect of HA-impaired conditioned medium relative to control conditioned medium to rescue growth in order to test whether HA is a necessary component of conditioned medium for growth rescue.

We thank the referees for this series of excellent suggestions, which we employed to substantiate the claim that HA is the relevant component of conditioned media mediating the rescue of GFAT1 KO cells, as follows.

HA is a polymer that can vary considerably in size, ranging from megadaltons to ~kilodaltons. During the course of this revision, we tested the hypothesis that GFAT1 knockout PDA cells survived and proliferated better when provided low molecular weight HA (known as oligo-HA, o-HA), as opposed to lager formulations. Contributing to this understanding, we previously demonstrated that HA digestion with hyaluronidase (HA-ase) promotes HA rescue of GFAT1 KO PDA cells (Figure 3G) and that 5kDa HA (i.e. o-HA) is more effective at rescuing proliferation than 60kDa HA (Figure 3H, Figure 5H-K, Figure 3—figure supplement 2A-D, Figure 5—figure supplement 1E-J). The rescue has also since been validated in a second cell line (Figure 3I, Figure 5—figure supplement 1G-J).

Bearing this in mind, we utilized a clinically established ELISA detection method to quantitate HA in CM from wild type PDA cells. This detection kit works extremely well for HA larger than ~50kDa, with a linear detection range extending into the megadalton range (Figure 3—figure supplement 2E). However, we found that enzymatic breakdown with HAase prevents detection by ELISA (Figure 3—figure supplement 2F,G). As expected, cancer associated fibroblasts (CAF) produced orders of magnitude more HA than did wild type PDA cells. And, while we were able to detect low levels of HA in CM from wild type PDA cells (Figure 4A, Figure 3—figure supplement 2F), this was well below the ~mM concentration required to provide rescue. Of note, the low level of HA detected in the CM by ELISA was degradable by HAase (Figure 3—figure supplement 2F,G). Collectively, these data added to our growing body of evidence that the rescue of GFAT1 knockout by PDA CM results from o-HA, which is not detectable by ELISA.

To test this hypothesis, we inhibited HA synthesis in wild type PDAC cells using the pan-HA synthase (HAS) inhibitor 4-Methylumbelliferone (4-MU) and measured HA in the CM (Figure 4A). The HA at baseline was again low; however, we were able to observe a decrease in HA concentration in a dose-dependent fashion, illustrating on-target activity of 4-MU. More importantly, we observed that 4-MU treatment of wild type PDAC cells during CM preparation decreased the rescue activity of the associated CM in a dose dependent manner (Figure 4B-E), further suggesting that o-HA in the PDA CM is the relevant factor rescuing the growth of GFAT1 knockout cells. Finally, we also found that knockout of the GlcNAc salvage pathway by CRISPR/Cas9 (i.e. NAGK knockout) in GFAT1 knockout cells (GFAT1/NAGK double knockout) blocked the rescue of UDP-GlcNAc pools, proteome O-GlcNAcylation and proliferation by wild type PDA cell CM (Figure 5L,M and Figure 5—figure supplement 1A-D,K,N).

In sum, these data support the conclusion that wild type PDAC cells produce HA, which includes o-HA, and that this is the relevant molecule mediating the rescue of GFAT1 knockout PDA cells.

Absent these experiments, the authors should test whether GFAT1/NAGK double KO cells can form tumors in vivo in order to determine whether, besides HA cleavage and GlcNAc salvage via NAGK, there are alternative nutrients/pathways that may support glycosylation and cellular proliferation.

This is a terrific suggestion, and given the experiments detailed above, we hope the referees agree that this can be considered an area for future pursuit.

3. A related point centers around the notable context-specific environmental rescue of GFAT1 deletion. For example, Figures 1F and G show a complete rescue of GFAT1 KO in the subcutaneous setting, but a significant impairment of GFAT1 KO tumor growth in the orthotopic setting. How do HA levels compare in these distinct environments?

This is a great observation, and we sincerely appreciate the suggestion, which we believe may address the difference in degree of rescue between these two anatomic tumor locations. Namely, we performed IHC for HA binding protein (HABP), a well-known marker for HA in tumors/tissues (e.g. Provenzano, et al., *Cancer Cell* 2012; Gebauer, et al., *PLOS One* 2017), in sections from WT and GFAT1 knockout subcutaneous and orthotopic tumors. This was compared to staining of normal pancreas (negative control) and an autochthonous murine pancreatic tumor (positive control) (Figure 4F). Blinded scoring of 10 sections per sample using a 0-3 scale (Figure 4G) revealed considerably greater HABP staining in the subcutaneous tumors (Figure 4H). Additionally, while the HABP content in subcutaneous tumors was independent of tumor genotype (i.e. wild type vs GFAT1 knockout), there was a significant decrease in HABP content in the GFAT1 knockout orthotopic tumors, consistent with the decreased tumor growth (Figure 1G).

Gebauer F, Kemper M, Sauter G, Prehm P, Schumacher U. Is hyaluronan deposition in the stroma of pancreatic ductal adenocarcinoma of prognostic significance? PLoS One. 2017 Jun 5;12(6):e0178703. doi: 10.1371/journal.pone.0178703. PMID: 28582436; PMCID: PMC5459453.

Provenzano PP, Cuevas C, Chang AE, Goel VK, Von Hoff DD, Hingorani SR. Enzymatic targeting of the stroma ablates physical barriers to treatment of pancreatic ductal adenocarcinoma. Cancer Cell. 2012 Mar 20;21(3):418-29. doi: 10.1016/j.ccr.2012.01.007. PMID: 22439937; PMCID: PMC3371414.

Similarly, the conditioned medium rescue data presented in Figure 2, particularly the increased rescue by PDAC cell conditioned medium compared to CAF conditioned medium, is difficult to interpret together with the in vivo results in Figure 1F and G which suggest that the mechanism underlying rescue of proliferation is independent of PDAC cells themselves. Do GFAT1-replete PDAC cells secrete HA, and if so, do the levels exceed HA in CAF conditioned medium? To address these questions, the authors could perform ELISA-based assays for HA present in conditioned medium and IHC for HA binding protein in vivo.

We appreciate this suggestion too, the response for which we will build off response #2 above. By utilizing the HA ELISA described, we found that CAF CM contains much more HA than does wild type PDA CM (Figure 3—figure supplement 2F). This HA is high molecular weight, based on the fact that it can be detected by the ELISA, which can be eliminated by treatment with HAase (Figure 3—figure supplement 2F). In other words, the HA in CAF CM (high molecular weight) parallels the lower rescue activity in GFAT1 knockout proliferation assays. Reciprocally, the HA composition of the PDA CM (low molecular weight/o-HA) parallels the higher rescue activity.

In sum, the (i) HA composition of the CAF vs PDA CM (Figure 3—figure supplement 2F,G); together with the (ii) data using HA of varying sizes (Figure 3H,I, Figure 5H-K, Figure 3—figure supplement 2C,D, Figure 5—figure supplement 1E-J); the (iii) influence of HAase on HA-mediated rescue (Figure 3G); the (iv) observation that we can impair the rescue activity of wild type PDA CM by inhibiting HAS (Figure 4A-E); and the rescue of UDP-GlcNAc pools and the O-GlcNAcylated proteome in GFAT1 knockout, but not GFAT1/NAGK double knockout (Figure 5L,M, Figure 5—figure supplement 1K,N), collectively form the basis for our conclusion that o-HA in PDA CM is responsible for the rescue of GFAT1 knockout cells. While we believe that our data are convincing, we did not definitely detect/quantitate o-HA in PDA CM, and we are careful to make note of this in our revised manuscript.

Regarding the latter point, we performed IHC for HABP in tumor sections from GFAT1 proficient and GFAT1 deficient tumors (Figure 4F,G). The similarity in HABP content between GFAT proficient versus deficient subcutaneous tumors suggests that the HA was deposited by another cell type, e.g. CAFs. Indeed, it was not our intention to indicate that the HA in PDA tumors was deposited by the cancer cells. Rather, we found that wild type PDA CM could rescue the proliferation of GFAT1 knockout PDA cells in vitro, and we utilized this as a tool to study the process. In the revised manuscript we are careful to make this distinction, as we neither aimed to determine, nor did our data definitively conclude, what cell types in the pancreatic TME produce/deposit HA.

4. To illustrate the generalizability of their model, the authors should repeat key findings, including the ability of hyaluronic acid to rescue growth of cells cultured in low nutrients or with GFAT1 deletion, in at least 2 of the cell lines generated and used in the manuscript. Similarly, the results of HA supplementation in GFAT1/NAGK double KO cells (Figure 4H-M) lends support to the importance of hexosamine salvage for PDAC cell proliferation, but this is only shown in the 8988T cell line and should be reproduced in an additional PDAC cell line.

We absolutely agree with this important point. In our revised study, the key findings noted have been repeated in at least 2 cell lines: HA Uptake (Figure 3F); wild type PDA CM and HA rescue of GFAT1 knockout, including the impact of 60kDA vs 5kDA HA (Figure 3H,I, Figure 3—figure supplement 2A,B, Figure 5—figure supplement 1G,H); and block upon knockout of NAGK in GFAT1 knockout cells of HA or wild type PDA CM-mediated rescue of proliferation (Figure 5H-K, Figure 5—figure supplement 1A-J) and proteome O-GlcNAcylation (Figure 5L, Figure 5—figure supplement 1N).

The results from experiments aimed to enhance growth of wild type PDA cells in low nutrient media with HA have proven challenging to consistently reproduce. This assay appears sensitive to plating conditions and media/reagent formulation. We are working to delineate the relevant variables, and given this lack of confidence in the data, we opted to remove these data from the revised manuscript. We are continuing to work on this line of investigation and hope the referees agree that this can be considered an area for future pursuit.

Title:We recommend the authors change the title to "Hyaluronic Acid Fuels Pancreatic Cancer Cell Growth" to reflect the fact that experiments testing the role of hyaluronic acid supporting cancer cell growth was performed in vitro.

We agree that this title more accurately reflects our findings. We have updated it accordingly.